# VPS38/UVRAG and ATG14, the variant regulatory subunits of the ATG6/Beclin1-PI3K complexes, are crucial for the biogenesis of the yolk organelles and are transcriptionally regulated in the oocytes of the vector *Rhodnius prolixus*

**Priscila H. Vieira[1], Claudia F. Benjamim**  **[2], Georgia Atella** [3,4]**, Isabela Ramos** [1,4]*

**1** Laboratório de Bioquímica de Insetos, Instituto de Bioquímica Médica Leopoldo de Meis. Universidade Federal do Rio de Janeiro, Brazil, **2** Laboratório de Imunologia Molecular e Celular, Instituto de Biofísica Carlos Chagas Filho (IBCCF), Universidade Federal do Rio de Janeiro, Brazil, **3** Laboratório de de Bioquímica de Lipídeos e Lipoproteínas, Instituto de Bioquímica Médica Leopoldo de Meis. Universidade Federal do Rio de Janeiro, Brazil, **4** Instituto Nacional de Ciência e Tecnologia em Entomologia Molecular–INCT-EM/CNPq. Rio de Janeiro, Brazil

* isabela@bioqmed.ufrj.br

## Abstract

In insects the reserve proteins are stored in the oocytes into endocytic-originated vesicles named yolk organelles. VPS38/UVRAG and ATG14 are the variant regulatory subunits of two class-III ATG6/Beclin1 PI3K complexes that regulate the recruitment of the endocytic (complex II) and autophagic (complex I) machineries. In a previous work from our group, we found that the silencing of ATG6/Beclin1 resulted in the formation of yolk-deficient oocytes due to defects in the endocytosis of the yolk proteins. Because ATG6/Beclin1 is present in the two above-described PI3K complexes, we could not identify the contributions of each complex to the yolk defective phenotypes. To address this, here we investigated the role of the variant subunits VPS38/UVRAG (complex II, endocytosis) and ATG14 (complex I, autophagy) in the biogenesis of the yolk organelles in the insect vector of Chagas Disease *Rhodnius prolixus*. Interestingly, the silencing of both genes phenocopied the silencing of ATG6/Beclin1, generating 1) accumulation of yolk proteins in the hemolymph; 2) white, smaller, and yolk-deficient oocytes; 3) abnormal yolk organelles in the oocyte cortex; and 4) unviable F1 embryos. However, we found that the similar phenotypes were the result of a specific cross-silencing effect among the PI3K subunits where the silencing of VPS38/UVRAG and ATG6/Beclin1 resulted in the specific silencing of each other, whereas the silencing of ATG14 triggered the silencing of all three PI3K components. Because the silencing of VPS38/UVRAG and ATG6/Beclin1 reproduced the yolk-deficiency phenotypes without the cross silencing of ATG14, we concluded that the VPS38/UVRAG PI3K complex II was the major contributor to the previously observed phenotypes in silenced insects.

**Data Availability Statement:** All relevant data are within the manuscript and its Supporting Information files.

**Funding:** This work was funded by the following grants. Fundação Carlos Chagas Filho De Amparo À Pesquisa Do Estado Do Rio De Janeiro (FAPERJ) (JCNE E-26/2031802017; http://www.faperj.br/) to I.R.; Conselho Nacional de Desenvolvimento Científico e Tecnológico (CNPq) (INCT-EM 16/2014; http://cnpq.br/) to I.R.; Coordenação de Aperfeiçoamento de Pessoal de Nível Superior (CAPES) (https://www.gov.br/capes/pt-br) to I.R. The funders had no role in study design, data collection and analysis, decision to publish, or preparation of the manuscript.

**Competing interests:** The authors have declared that no competing interests exist.

Altogether, we found that class-III ATG6/Beclin1 PI3K complex II (VPS38/UVRAG) is essential for the yolk endocytosis and that the subunits of both complexes are under an unknown transcriptional regulatory system.

## Author summary

The oocytes of oviparous animals are highly specialized cells committed to the storage of nutrients (yolk) required for the maternally detached embryo development. Most oocytes undergo a massive growth during oogenesis due to their high endocytic activity of yolk macromolecules. In this work, we found that the components of two PI3K complexes are co-transcriptionally regulated, and that one specific PI3K complex (class-III PI3K complex II) is essential for the correct endocytosis and biogenesis of the yolk endocytic organelles. The molecular machinery and regulations that govern the yolk generation are mostly unknown. Thus, these findings are important in the context of general vector's reproductive biology and for the elaboration of novel strategies for vector's population control.

## Introduction

The oocytes of oviparous animals are highly specialized cells dedicated to the storage of nutrients required for embryo development. Most oocytes undergo a massive growth in volume during oogenesis due to the endocytosis of yolk macromolecules which will be targeted for degradation in a regulated manner throughout embryogenesis [1–5].

Phosphoinositide 3-kinase (PI3K) family members are able to generate phosphatidylinositol-3P (PI3P), an important signal molecule involved in a variety of vesicle trafficking events, in a spatially- and timely-regulated manner. The controlled generation of this important signal inducer regulates diverse cellular processes including cell growth, proliferation, and survival [6–9]. Three classes of PI3K have been described, and each class includes several different PI3K complexes. Typically, PI3P produced by class-III PI3K complexes have been found in autophagosomes, endosomes and multivesicular bodies [6, 10–14]. Here, we focused on the role of the two autophagy related gene 6 (ATG6/Beclin1) class-III PI3K complexes, which are involved in the regulation of autophagy and the membrane dynamics of endocytosis [15].

ATG6/Beclin1 is an autophagy related gene component of two different class-III PI3K complexes. The invariant subunits of both complexes, in addition to ATG6/Beclin1, are the catalytic subunit vacuolar protein sorting 34 (VPS34) and the putative protein kinase vacuolar protein sorting 15 (VPS15), which were originally described in *Saccharomyces cerevisiae* but have been found in other species including plants and higher eucaryotes [16]. The distinct cellular functions of each ATG6/Beclin1 PI3K complex is related to the association of these complexes to different regulatory subunits vacuolar protein sorting 38/ UV irradiation resistance-associated gene (VPS38/UVRAG) or autophagy related gene 14 (ATG14). The complex I is composed by ATG6/Beclin1, VPS34, VPS15 and ATG14. It is formed on the pre-autophagosome structure and is involved in the recruitment of ATGs and regulation of autophagy initiation. The complex II is composed by ATG6/Beclin1, VPS34, VPS15 and VPS38/UVRAG. This complex is localized in the membrane of endosomes, being important to general vacuolar protein sorting [12, 17]. In brief, ATG14 is a key subunit in determining the role of the PI3K complex during autophagy, while VPS38 is essential to mobilize the vesicles along the endocytic

route [15, 18, 19]. However, recent studies have demonstrated that not only the ATGs are involved in the regulation of other cellular functions, such as programmed cell death, aging, immune system, viral replication, tumorigenesis and others [20–33], but also that VPS38/ UVRAG plays alternative functions that became more evident in the past few years. For example, VPS38/UVRAG participates in the regulation of the translocation of autophagy related gene 9 (ATG9) (a lipid scramblase that mediates autophagosomal membrane expansion [34]), to the autophagosome and its association with important effectors of retrograde vesicular transport [35, 36].

The participation of receptor mediated endocytosis in the yolk uptake and biogenesis of the yolk organelles have been investigated in oocytes of many species. In insects, the internalization of yolk proteins through the presence of a specialized endocytic cortex in the oocytes, which includes prominent microvilli, coated pits, coated vesicles, and endosomes have been shown in several species [37–44]. Immunodetection of the internalization of yolk proteins and markers of endocytosis (such as clathrin, α-adaptin, and yolk protein receptors) have been performed in Drosophila [44] and mosquito [45]. However, the regulations encompassing the recruitment of the endocytic machinery to specific sites of the oocyte cortex and the signals and regulations that govern the oocyte endocytic pathways and endosomal maturation have never been addressed.

In a previous work our group demonstrated that ATG6/Beclin1 is important for the uptake of yolk proteins, yolk organelles biogenesis and embryo viability in the insect vector *R. prolixus* [46]. Here, we investigate the role of the ATG6/Beclin1-PI3K complexes variant regulatory subunits: ATG14 and VPS38/UVRAG. We found that for both VPS38/UVRAG- and ATG14-silenced females, the yolk proteins are accumulated in the hemolymph as the result of poor uptake by the oocytes. As a consequence, the silenced mature oocytes are small, white (as opposed to control oocytes which are light red), present abnormal yolk organelles, as seen by light microscopy and flow cytometry, and do not support F1 embryo development. Because we learned that VPS38/UVRAG and ATG14 remarkably phenocopied each other and the effects of silencing ATG6/Beclin1 [46], we asked if there was a transcriptional regulation among these genes. Indeed, we found a specific cross-silencing among VPS38/UVRAG, ATG14 and ATG6/Beclin1 in all RNAi knockdown ovary samples, which explains the phenocopies and reveals the existence of transcriptional regulations among the subunits of PI3K complexes.

## Material and methods

### Ethics statement

All animal care and experimental protocols were approved by guidelines of the institutional care and use committee (Committee for Evaluation of Animal Use for Research from the Federal University of Rio de Janeiro, CEUA-UFRJ #01200.001568/2013-87, order number 155/13), under the regulation of the national council of animal experimentation control (CONCEA). Technicians dedicated to the animal facility conducted all aspects related to animal care under strict guidelines to ensure careful and consistent animal handling.

### Bioinformatics

The sequence of *Rhodnius prolixus* VPS38/UVRAG (RPRC001388), ATG14 (RPRC001958) and ATG6/Beclin1 [46] were obtained from the *R. prolixus* genome and transcriptome databases (Rpro C3.2) from Vector Base (www.vectorbase.org) [47–49]. The Drosophila VPS38/ UVRAG (Gene ID: 34735) and ATG14 (Gene ID: 43438) orthologues were used as the first template for identification. Values of similarity and identity were predicted using SIAS

software (www.imed.med.ucm.es/Tools/sias.html) and compared to the sequences of *Homo Sapiens* UVRAG/VPS38 (Gene ID: 7405) and ATG14 (Gene ID: 2286) and *Saccharomyces cerevisiae* UVRAG/VPS38 (Gene ID: 851074) and ATG14 (Gene ID: 852425). Conserved domains were predicted using PFAM (http://pfam.xfam.org/) [50].

## Insects

Insects were maintained at a 28 ± 2˚C controlled temperature, relative humidity of 70–80% and 12/12 h light and dark cycles. All females used in this work were obtained from our own insectarium where mated females are fed for the first time (as adult insects) in live-rabbit blood 14 to 21 days after the 5th instar nymph to adult ecdysis. After the first blood feeding, all adult insects in our insectarium are fed every 21 days. For all experiments, mated females of the second or third blood feeding were used, and dissections were carried out at different days after the blood meal depending on the experiment.

## Dissection of the ovary parts, follicles, and chorionated oocytes

The different parts of the ovariole were carefully dissected in phosphate buffered saline (PBS) 137 mM NaCl, 2.7 mM KCl, 10 mM $Na_2HPO_4$, and 1.8 mM $KH_2PO_4$, pH 7.4 using fine tweezers and dissecting scissors under the stereo microscope according to [51], and the structures (tropharium, previtellogenic and vitellogenic follicles, and chorionated oocytes) were classified by length and morphology according to [52, 53]. The chorionated oocytes were dissected from the ovarioles (not from the oviducts). Because it is not known when oocyte maturation or activation occurs in *R. prolixus*, there is no way of knowing if the chorionated oocyte is still an immature oocyte or a mature egg. The terminology chorionated oocyte (and not chorionated egg) was chosen to refer to the oocytes right after the completion of choriogenesis because it matches to the terminology used in Drosophila [54, 55]. When chorionated oocytes were used, dissection was performed 7–12 days after the blood meal. All other organs and ovary parts were dissected 7 days after blood meal.

## Extraction of RNA and cDNA synthesis

All samples were homogenized in Trizol reagent (Invitrogen) for total RNA extraction. Reverse transcription reaction was carried out using the High-Capacity cDNA Reverse Transcription Kit (Applied Biosystems) using 1 μg of total RNA after RNase-free DNase I (Invitrogen) treatment, Multiscribe Reverse Transcriptase enzyme (2.5 U/μL) and random primers for 10 min at 25˚C followed by 2 hours of incubation at 37˚C. As a control for the DNAse treatment efficiency, we performed control reactions without the enzyme followed by testing the capacity of amplification by PCR.

## PCR/RT-qPCR

Specific primers for the RpVPS38 and RpATG14 sequences were designed to amplify 323bp and 219bp fragments, respectively, in a PCR using the following cycling parameters: 10 min at 95˚C, followed by 35 cycles of 15 s at 95˚C, 45 s at 52˚C and 30 s at 72˚C and a final extension of 15 min at 72˚C. Amplifications were observed in 2% agarose gels. Quantitative PCR (qPCR) was performed in a StepOne Real-Time PCR System (Applied Biosystems) using SYBR Green PCR Master Mix (Applied Biosystems) under the following conditions: 10 min at 95˚C, followed by 40 cycles of 15 s at 95˚C and 45 s at 60˚C. qPCR amplification was performed using the specific primers described in S1 Table. All primers were used at a final concentration of 0.2 μM. The cDNAs were diluted 10X and used in the reactions. To exclude nonspecific

amplification, blank reactions replacing de template (cDNA) for water were performed in all experiments. The relative expressions were calculated using the delta $C_t$ (cycle threshold) obtained using the reference gene 18S (RPRC017412) and calculated $2^{-dCt}$. According to the minimum information for publication of quantitative RT-qPCR experiments (MIQE) Guidelines, normalization against a single reference gene is acceptable when the investigators present clear evidence that confirms its invariant expression under the experimental conditions [56]. S1 Fig shows the invariant expression of 18S in our experimental conditions. The primers for VPS38/UVRAG presented 84.52% efficiency with a slope of -3.759 and 25.01 Y-interception. The primers for ATG14 presented 100% efficiency with a slope of -3.28 and 25.872 Y-interception. For all RT-qPCRs, each biological replicate (n = 6–8) was prepared using a pool of samples dissected from 3 insects.

### RNAi silencing

dsRNA was synthesized by MEGAScript RNAi Kit (Ambion Inc) using primers for RpVPS38/UVRAG and RpATG14 specific gene amplification with the T7 promoter sequence (S1 Table). dsRNA for the silencing of ATG6/Beclin1 was synthesized as previously described [46]. Unfed adult females were injected between the second and third thoracic segments using a 10 μl Hamilton syringe with 1 μg dsRNA (diluted in 1μl of water) and fed 2 days later. Knockdown efficiency was confirmed by qPCR at 7 days after blood meal in the ovary and fat body. The bacterial *MalE* gene was used as a control dsRNA [20]. Adult females injected with dsRNA were fed and transferred to individual vials. The mortality rates and the number of eggs laid by each individual were recorded daily and weekly, respectively. Additional measurements are described below.

### PI3P detection by thin-layer chromatography (TLC) and quantification

Silenced and control females were fed with blood enriched with 32Pi [57] using a special feeder [58]. On the tenth day after a blood meal, the chorionated oocytes were collected and subjected to lipid extraction [59]. The thin-layer silica plates (G-60; 0±25 mm thickness) were pre-coated with trans-1,2-diaminocyclohexane-N,N,N',N'- tetraacetic acid solution in water: methanol: 10M NaOH (1:2:0,1 v/v) and allowed to dry overnight [60]. The TLC developing solution was prepared by stirring together methanol (75 ml), CHC13 (60 ml), pyridine (45 ml), and boric acid (12 g) until the boric acid was dissolved. The plates were developed in a saturated tank and were run to the top [60]. The radioactivity was analyzed in a laser scanner Cyclone1Plus Storage Phosphor System (Perkin Elmer). Additionally, the plates were autoradiographed using X-ray film (Kodak T-Mat) and an intensifier screen. The cassette was stored at −70°C for 30 days. The autoradiography was then subjected to densitometric analysis. The phospholipids spots were identified by comparison with standards run in parallel according to [61]. Densitometry was performed using ImageJ software. Each biological replicate (n = 5) was prepared using a pool of 2 chorionated oocytes.

### Egg homogenates and SDS-PAGE

Control and silenced eggs were collected at 24h of embryogenesis. Pools of 4 eggs were homogenized using a plastic pestle in 100 μl of PBS containing a cocktail of protease inhibitors (Aprotinin 0.3 μM, leupeptin 1 μg/μl, pepstatin 1 μg/μl, PMSF 100 μM and EDTA 1 mM). The samples were directly used, and 30 μg of total protein were loaded in each lane of a 13% SDS-PAGE. Gels were stained with silver nitrate [62]. For each biological replicate (n = 3), the samples were prepared using a pool of 4 eggs laid by different insects.

## Hemolymph extraction and SDS-PAGE

The hemolymph was extracted from control and silenced females, 7 days after the blood meal, as originally described by Masuda e Oliveira, 1985 [63]. Approximately 10 μl of hemolymph per female was obtained by cutting one of the insects' legs and applying gentle pressure to the abdomen. The hemolymph was collected using a 10μl pipette plastic tip. Once collected, the hemolymph was diluted 2x in PBS containing protease inhibitors (aprotinin 0.3 μM, leupeptin 1 μg/μl, pepstatin 1 μg/μl, PMSF 100 μM and EDTA 1 mM), and approximately 8 mg of phen-ylthiourea. The equivalent of 1 μl of hemolymph was loaded in each lane of a 15% SDS-PAGE. For each biological replicate (n = 4), the samples were prepared using a pool of 3 insects.

## Determination of protein content

The total amount of protein in the silenced and control eggs and hemolymph samples was measured by the Lowry (Folin) method, using as standard control 1–5 μg of BSA [64] in a E-MAX PLUS microplate reader (Molecular devices) using SoftMax Pro 7.0 as software. For each biological replicate of the hemolymph samples (n = 8), the samples were prepared using a pool of 3 insects. Each biological replicate of the egg homogenates (n = 6) were prepared using a pool of 4 eggs laid by different insects.

## Light microscopy

Opercula of recently dissected chorionated oocytes were carefully detached using a sharp razor blade under the stereomicroscope. The opercula-free oocytes were fixed by immersion in 4% freshly prepared formaldehyde in 0.1 M cacodylate buffer, pH 7.2 for 12 h at room tempera-ture. Samples were washed 3 times for 10 minutes in the same buffer and embedded in increas-ing concentrations (25%, 50%, 75% and 100%) of OCT compound medium (Tissue-TEK) plus 20% glucose as a cryoprotectant, for 12 h for each of the concentrations. Once infiltrated in pure OCT, 15–30 μm transversal sections of the chorionated oocytes were obtained in a cryo-stat. The slides were mounted in glycerol 50% followed by observation in a Zeiss Axio Imager D2 equipped a Zeiss Axio Cam MrM operated in a differential interferential contrast (DIC) mode. For each biological replicate (n = 3), one chorionated oocyte was dissected from a dif-ferent insect and processed as described above.

## Yolk organelles suspension and flow cytometry

Suspensions of yolk organelles were obtained by gently disrupting of recently dissected chorio-nated oocytes in PBS (2 oocytes in 250μl of buffer). The population profiles of the yolk organ-elles were acquired on a FACS Calibur instrument (BD Bioscience, USA) powered by CellQuest Pro software v5.1 and analyzed using Flowing Software 2.5.1. For each biological replicate (n = 5), the samples were prepared using a pool of 2 chorionated oocytes dissected from different insects.

## Statistics

The relative expression and $\Delta C_t$ values were calculated from obtained $C_t$ (cycle threshold) val-ues. The $C_t$ mean values obtained from the experiments were compared using One-way ANOVA followed by Tukey's multiple comparison test. Differences were considered signifi-cant at $P < 0.05$. The relative expression values ($2^{-\Delta Ct}$) were used only for graph construction. Other results were analyzed by Student's t-Test for the comparison of two different conditions and One-way ANOVA followed by Tukey's test for the comparison among more than two

conditions. Differences were considered significant at $p<0.05$. All statistical analyses were performed using the Prism 7.0 software (GraphPad Software).

## Results

### VPS38/UVRAG and ATG14 maternal RNAs are accumulated in the oocytes

In a previous work, our group described that silencing of ATG6/Beclin1 in vitellogenic females resulted in the production of white/small oocytes and eggs due to the abnormal uptake of yolk components from the hemolymph. ATG6/Beclin1 is part of two class-III PI3K complexes, that signal to different intracellular routes: endocytosis and autophagy. To further investigate each of the ATG6/Beclin1 PI3K complexes were contributing to the ATG6/Beclin1 phenotypes, we decided to investigate the silencing effects of VPS38/UVRAG and ATG14, the variant subunits present in each of the ATG6/Beclin1 PI3K complexes. We found one isoform of the genes VPS38/UVRAG (RPRC001388) and ATG14 (RPRC001958) in the *Rhodnius* genome assembly (RproC3), with a total of 8 and 3 exons, respectively (S2 Fig). VPS38/UVRAG and ATG14 predicted proteins are 70 and 80% similar to their human orthologs (S3 Fig), presenting their expected conserved domains (CC and C2, Pfam: PF04111 and PF00168) (Fig 1A and 1B). RT-qPCR showed that the ovary of *R. prolixus* expresses approximately 6x more VPS38/UVARG than the midgut, fat body and flight muscle (Fig 1C), whereas ATG14 is 5x more expressed in the ovary than in the flight muscle and midgut (Fig 1D). Within the ovary, the highest levels of VPS38/UVRAG and ATG14 mRNAs were detected in the tropharium, which expresses an average of 2x the mRNA levels of both genes detected in the developing follicles (pre-vitellogenic and vitellogenic follicles) and chorionated oocytes (Fig 1C and 1D).

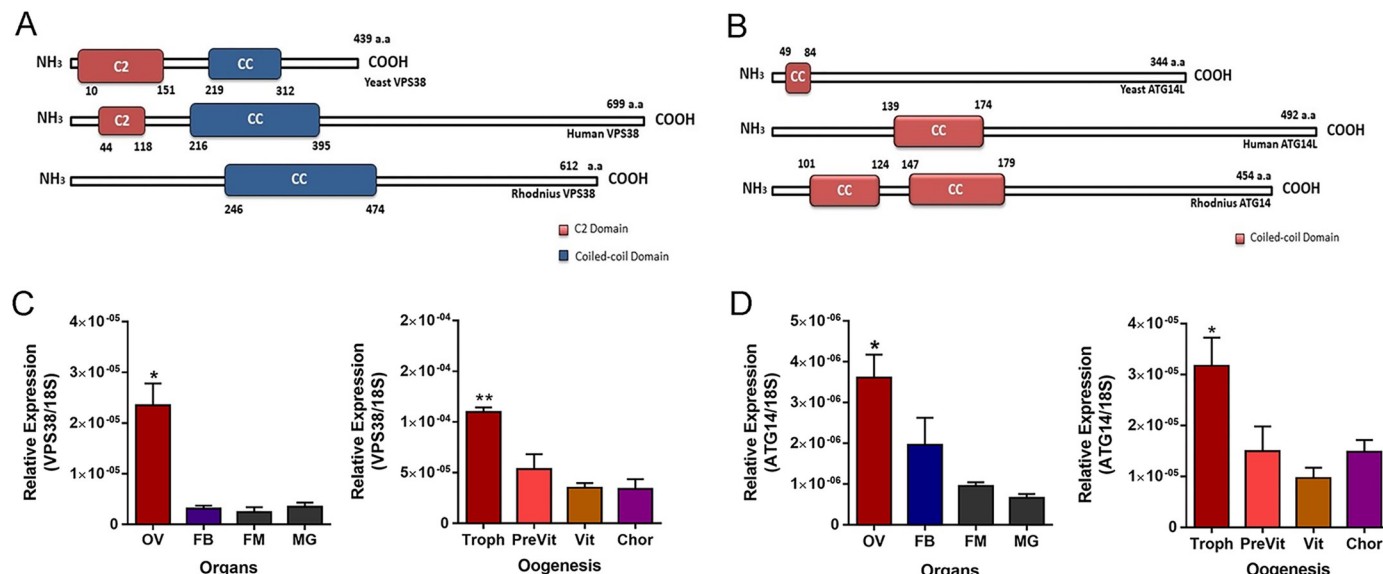

**Fig 1. VPS38/UVRAG and ATG14 are conserved and highly expressed in the ovaries. A-B.** Diagram of the predicted conserved functional domains of human, yeast and *Rhodnius prolixus* VPS38/UVRAG and ATG4. C2, a protein structural domain involved in targeting PI3K to the cell membrane, CC, coiled coil domain. **C-D.** RT-qPCR showing the relative expression of VPS38 and ATG4 in different organs and during oogenesis of vitellogenic females. MG, Midgut; FB, Fat body; FM, flight muscle; Ov, Ovary. Troph, tropharium; PreVit, pre-vitellogenic follicles; Vit, vitellogenic follicles; Chor, chorionated oocytes. The relative expression was quantified using the ΔCT method with Rp18S as endogenous control. Graphs show mean ± SEM (n = 6). * p<0.05, **p<0.01, One Way ANOVA.

## Silencing of VPS38/UVRAG and ATG14 results in the accumulation of yolk proteins in the hemolymph and formation of small yolk-deficient oocytes

Double stranded RNAs designed to target specific regions of the VPS38/UVRAG and ATG14 mRNAs were injected to the insects hemocoel 2 days before the blood meal. Nine days later (7 days after the blood meal), the ovaries and fat bodies of control and silenced females were dissected, and their mRNA contents were analyzed by RT-qPCR. The VPS38/UVRAG expression levels were reduced by 70% in the ovary and by 50% in the fat body (Fig 2A), whereas the ATG14 expression levels were decreased by 80% in the ovary and by 90% in the fat body (Fig 2B). Because PI3P is the product of PI3K complexes, we also tested the levels of PI3P in the ovaries of silenced females. We found that silenced ovaries presented a reduction of 60%, approximately, in their PI3P levels suggesting that their general PI3K activities were compromised (Figs 2C and S4). Despite the systemic knockdown efficiency, no alterations in the insect's main physiological characteristics such as digestion and longevity were observed in silenced animals when compared to control ones (S5 Fig).

When the silenced vitellogenic females were dissected, however, we observed that knockdown of both VPS38/UVRAG and ATG14 resulted in the production of small and white oocytes in the ovaries, remarkably similar to the pattern that we previously observed for the silencing of ATG6/Beclin1 [46] (Fig 3A and 3B). To test if the formation of the white/small oocytes was due to a deficiency in taking the yolk proteins from the hemolymph, as it happens in ATG6/Beclin1 insects [46], we quantified the hemolymph yolk protein content in control and silenced females. Higher concentrations of *Rhodnius heme binding protein* (RHBP) were effortlessly noticeable in the reddish hemolymph extracted from silenced females, when compared to the control female's transparent hemolymph (Fig 3C, detail image). RHBP is known for being the red molecule that provides the roseate color of *R. prolixus* oocytes and eggs [5, 65, 66]. For both VPS38/UVRAG and ATG14, the overall levels of protein in the hemolymph were increased by 30% (Fig 3C), and the fragments compatible with the major yolk proteins Vitellogenin [63] (Vg) (arrowheads) and RHBP [5, 65] were accumulated in the hemolymph (Fig 3D).

## VPS38/UVRAG and ATG14 silenced eggs are yolk-deficient and do not support embryo development

Despite the reduced size and abnormal color of the VPS38/UVRAG and ATG14 silenced oocytes (Fig 3B), the silenced females laid similar number of eggs as control females (Fig 4A and 4B). Fig 4C shows the F1 phenotypic distribution for control and silenced females. For VPS38/UVRAG an average of 11 eggs/ female (22% of the total number of eggs) were white/small, and for ATG14 an average of 10 eggs/female (27% of the total number of eggs) presented the white/small phenotype (Fig 4C). All eggs presenting the white/small phenotype (Fig 4D) were not viable (0% hatching rate), whereas the apparently morphologically normal eggs laid by silenced females decreased their viability to 67% and 55% for VPS38/UVRAG and ATG4, respectively (Fig 4C, numbers inside the bars) (S6 Fig).

VPS38/UVRAG and ATG14 white/small eggs (Fig 4D) accumulated only 15% and 20%, respectively, of the total protein found in the eggs produced by control females (Fig 4E), and the accumulation of the main yolk proteins, vitellin (arrowheads) and RHBP (arrow), was severely compromised as seen by SDS-PAGE (Fig 4F). The approximated predicted volumes of the silenced white eggs are 50–60% smaller than control eggs, with an average volume of $0.82 \pm 0.01 mm^3$ for ATG14 and of $0.67 \pm 0.03 mm^3$ for VPS38/UVRAG versus $1.58 \pm 0.04 mm^3$ of the eggs laid by control females.

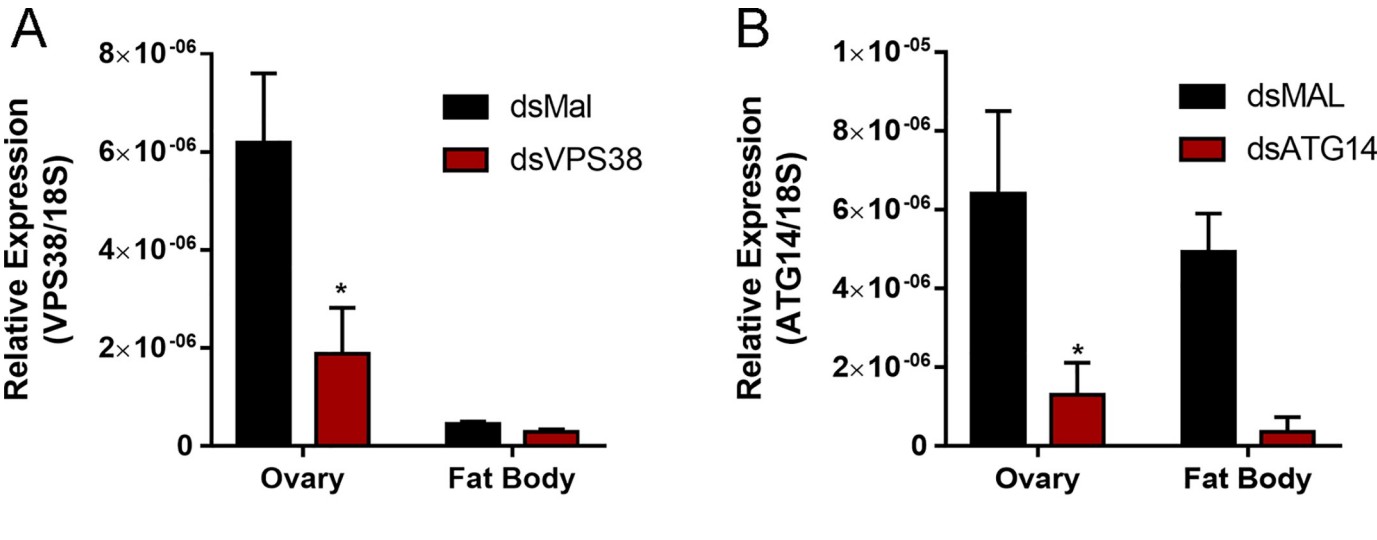

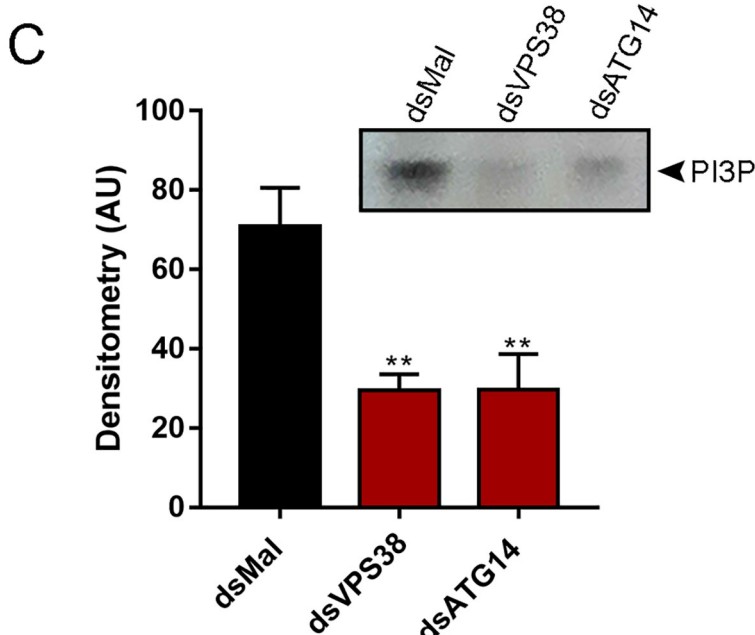

**Fig 2. RNAi knockdown of VPS38/UVRAG and ATG14 results in reduced levels of PI3P in the ovary. A-B.** RT-qPCR showing the relative expression levels of VPS38/UVRAG and ATG14 in the ovary and fat body of control (dsMal) and silenced females. The organs were dissected 7 days after the blood meal (n = 8). The expression levels of VPS38/UVRAG were reduced by 70% in the ovary and by 50% in the fat body. The expression levels of ATG14 were reduced by 80% in the ovary and by 90% in the fat body. **C.** Densitometric measurements of PI3P detection by TLC in the chorionated oocytes of control and silenced insects (n = 5). * $p < 0.05$, ** $p < 0.01$, One Way ANOVA.

## VPS38/UVRAG and ATG14 are important for the biogenesis of the yolk organelles in the oocyte cortex

Transversal cryosections of chorionated oocytes showed an accumulation of larger yolk organelles in the core cytoplasm (red asterisk) and an irregular distribution of smaller and abnormal organelles in the periphery of the oocytes (white arrowhead) (Fig 5A). This morphology is similar to what we previously observed in ATG6/Beclin1-silenced oocytes and eggs [46] and

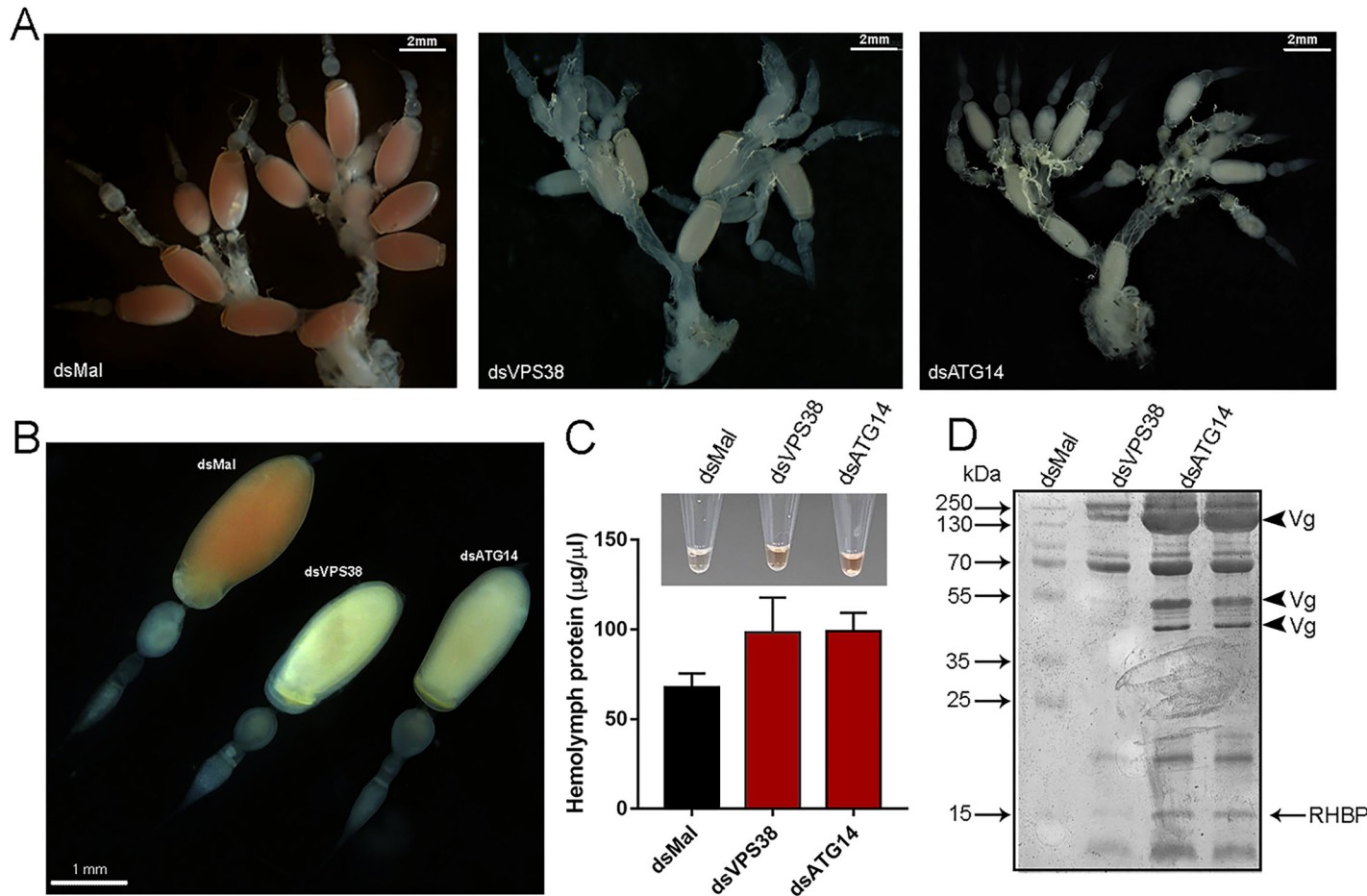

**Fig 3. Silencing of VPS38/UVRAG and ATG14 results in small and white oocytes. A.** Representative images of the ovaries of vitellogenic females injected with dsMal, dsVPS38/UVRAG and dsATG14. **B.** Representative image of ovarioles dissected from females previously injected with dsMal, dsVPS38/UVRAG and dsATG14. **C.** Image: representative image of the hemolymph samples extracted from females previously injected with dsMal, dsVPS38/UVRAG and dsATG14. Graph: total amount of protein in the hemolymph from silenced and control females. Graph shows mean ± SEM (n = 8). **D.** 13% SDS-PAGE showing the protein profile of the hemolymph from silenced and control females. All experiments were performed 7 days after the blood meal. Arrowheads, Vg, vitellogenin subunits; RHPB, *Rhodnius* heme binding protein (n = 4).

suggests problems in the biogenesis/ sorting of the yolk organelles, compatible with the hypothesis that silenced oocytes were not able to properly recruit the endocytosis machinery during oogenesis. To further investigate the characteristics of the yolk organelles formed in the oocytes of silenced females, we used flow cytometry to get further insight into the changes in the population of organelles in terms of size (FSC) and internal complexity (SCC), which is an unprecedent attempt. Isolation of the yolk organelles in *R. prolixus* have been performed before and the vesicles can vary in size from 500 nm to 40 μm [67, 68]. Fig 5B shows representative dot-plots of the yolk organelles present in chorionated oocytes from control and silenced females. The representative plots allowed the identification of a broad range of events and showed that although the organelles are dispersed over a large range of sizes and internal complexities, the organelles from control and silenced samples presented different profile as quantified in Fig 5C. For both VPS38/UVRAG and ATG14, there is an increase in frequency of the organelles in the left bottom quadrant (LB), indicating reduced complexity and size in the general population of the yolk organelles obtained from silenced samples. Interestingly, for ATG14 the high complex and smaller organelles are decreased in frequency (left top quadrant,

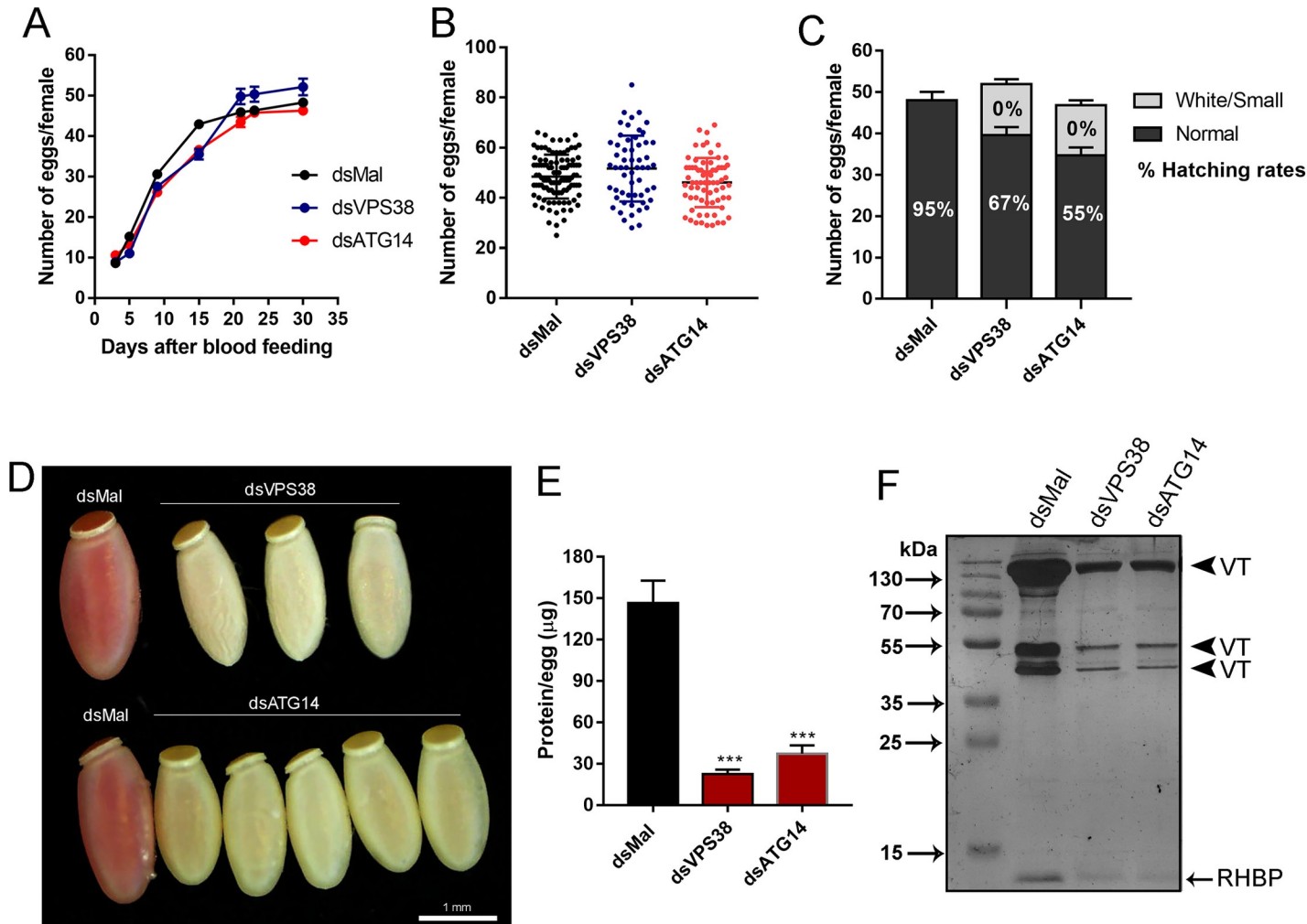

**Fig 4. Silencing of VPS38/UVRAG and ATG14 does not alter oviposition but decreases embryo viability. A.** Number of eggs laid per female over 4 weeks after the blood meal. Graph shows mean ± SEM (n = 52). **B.** Total of eggs laid by silenced and control females. Graph shows mean ± SEM (n = 52). **C.** Phenotypic distribution of the F1 eggs observed after knockdown of VPS38/UVRAG and ATG14. Percentage (%) of hatching per phenotype is also showed inside the bars. **D.** Representative image of the phenotypes observed in the F1 eggs laid by control and silenced females. **E.** Total amount of protein from control and VPS38/UVRAG and ATG14 silenced eggs, measured by the Lowry method. Graph shows mean ± SEM (n = 6). ***p<0.001, One Way ANOVA. **F.** 13% SDS-PAGE showing the protein profile of freshly laid eggs from control and silenced females. Arrowheads, VT, Vitellin subunits; RHPB–*Rhodnius* Heme Binding Protein (n = 3).

LT), whereas for VPS38/UVRAG the low complex and larger organelles are less frequent (right bottom quadrant, RB).

## VPS38/UVRAG, ATG14 and ATG6/Beclin1 are transcriptionally regulated

Because the phenotypes of both VPS38/UVRAG and ATG14 were remarkably similar to each other and to the phenotypes of silencing ATG6/Beclin1 [46], we decided to test if the RNAi silencing of one of those genes was, somehow, interfering in the expression of the other genes. Interestingly, we found that the silencing of VPS38/UVRAG and ATG6/Beclin1, resulted in the silencing of both VPS38 and ATG6, and did not trigger the silencing of ATG14, whereas the silencing of ATG14 resulted in the silencing ATG14, VPS38/UVRAG and ATG6/beclin1 (Fig 6). As a control we tested the expression levels of other genes, non-related to the class III PI3K complexes, such as ATG8/LC3, ATG1/ULK1, the elongation factor 1-alpha (EF1) and

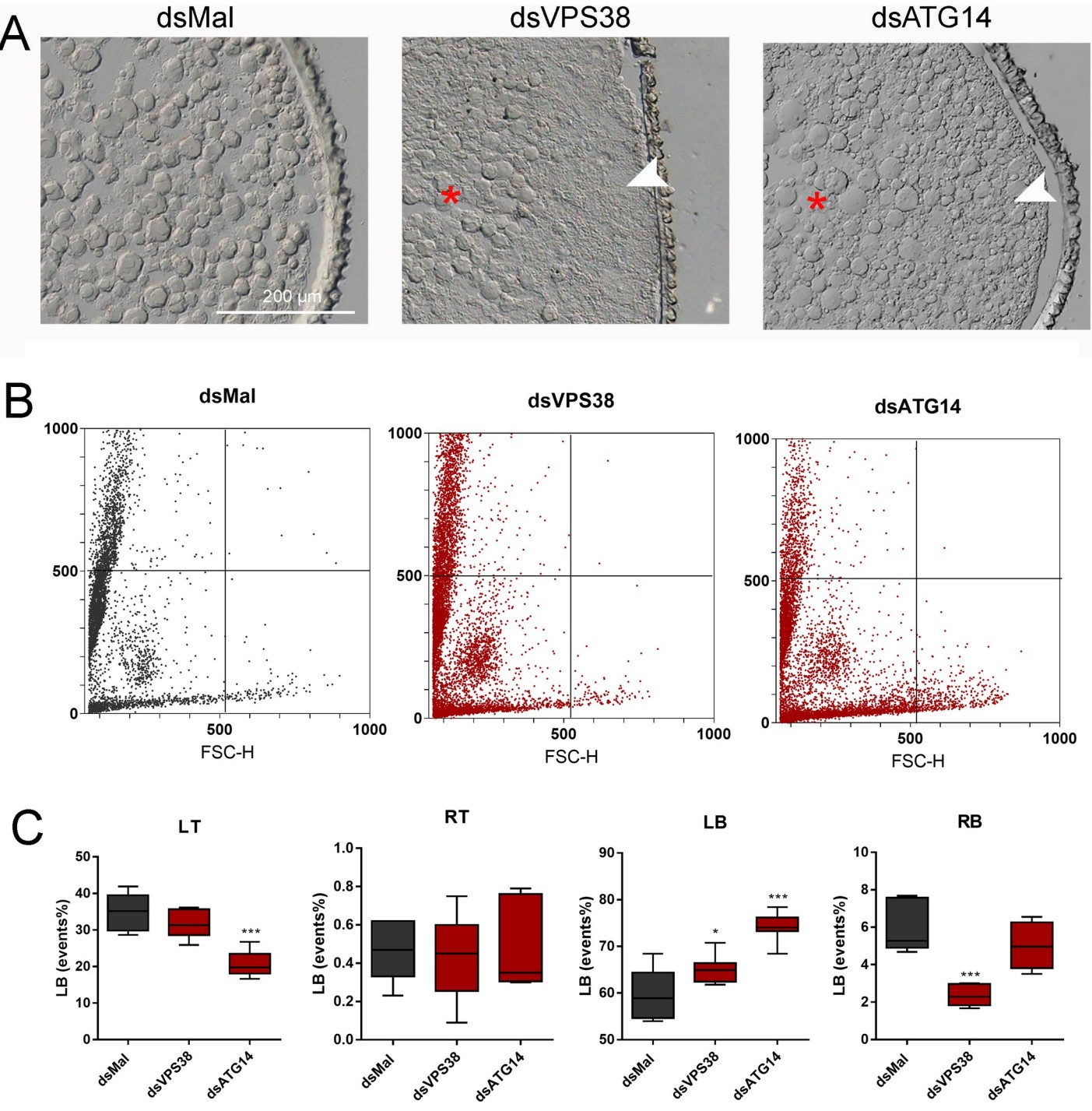

**Fig 5. Knockdown of VPS38/UVRAG and ATG14 results in abnormal distribution of the yolk organelles. A.** Representative images of cross-sections from chorionated oocytes were observed in the light microscope. The images show an accumulation of larger yolk organelles in the core cytoplasm (red asterisk) and an irregular distribution of smaller/abnormal organelles in the periphery of silenced oocytes (white arrowheads). Bars: 200 μm. (n = 3). **B.** Flow cytometry FSC x SSC dot-plots of the yolk organelles extracted from chorionated oocytes obtained from control and silenced females. The plots are representative of five experiments (n = 5). **C.** Quantification of the organelles (events) frequency in each quadrant of the plots shown in **B** (RT, Right top; LT, left top; RB, right bottom; LB, left bottom). (n = 5) *p<0.05,***p<0.001, One Way ANOVA.

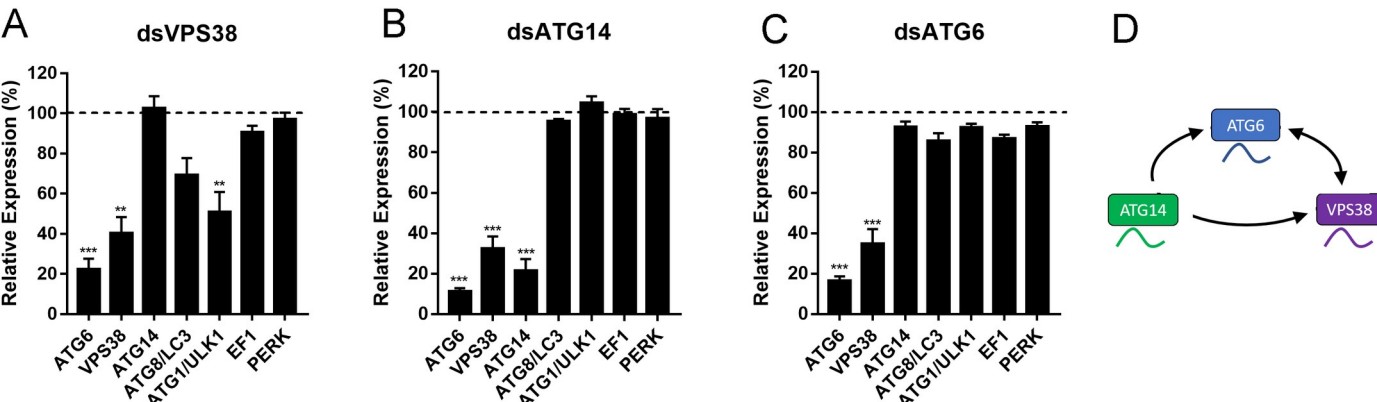

**Fig 6. VPS38/UVRAG, ATG14 and ATG6/Beclin1 are transcriptionally regulated in the oocytes of *Rhodnius prolixus*.** Oocytes dissected from silenced females were tested for the expression levels of the other class III PI3K components, and for the control genes ATG8/LC3, ATG1/ULK1, EF1 and PERK (non-related to the class III PI3K complexes). **A.** Expression levels of different genes in the ovaries of females silenced for VPS38/UVRAG (dsVPS38). **B.** Expression levels of different genes in the ovaries of females silenced for ATG14 (dsATG14). **C.** Expression levels of different genes in the ovaries of females silenced for ATG6/Beclin1 (dsATG6/Beclin1). Graphs show mean ± SEM. (n = 8) **p<0.01,***p<0.001. One Way ANOVA. **D.** Model for the transcriptional regulations between ATG6/Beclin1, VPS38/UVRAG and ATG14.

the unfolded protein response sensor PKR-like ER kinase (PERK). We found that, with the exception of ULK1/ATG1 in VPS38/UVRAG-silenced samples, which was also moderately silenced, PI3K non-related genes showed no significant changes in expression levels in the silenced samples for VPS38/UVRAG, ATG14 and ATG6/Beclin1 (Fig 6).

## Discussion

In 2005, the WHO created the department of Neglected Tropical Diseases (NTDs), recognizing their importance and aiming to manage their incidence mostly in Africa and Latin America (http://www.who.int/neglected_diseases/en/). *Rhodnius prolixus* is one of the main vectors of Chagas disease, which is one of the several NTDs that are important in Latin America. Currently 6 to million people are estimated to be infected by Chagas Disease, and vector control is still the most useful method to prevent this illness. Because embryo development relies entirely on the buildup of the yolk nutritional reserves, the proper biogenesis of the yolk organelles is indispensable for vector reproduction representing a promising target for interference in vector's reproduction.

The critical roles of the ATG6/Beclin1-associated complexes in the endocytic trafficking and autophagy pathways have been described before in models such as Drosophila and *C. elegans* [69–72]. Interestingly, in the hard tick *Haemaphysalis longicornis* depletion of ATG6/Beclin1 also results in a repression of Vg uptake by the oocytes [73] and in *Caenorhabditis elegans*, disruption of ATG6/Beclin1 results in defects in the secretion and localization of the lipoprotein receptor-related protein 1 (LRP1) [74]. Both LRP1 and Vg are important for oogenesis and their disruption might also be the result impaired endocytic activities in those models.

Similar to the results previously obtained by our group with ATG6/Beclin1 [46], we found that the systemic silencing of both VPS38/UVRAG and ATG14 resulted in specific phenotypes of impaired yolk uptake in vitellogenic females. This is compatible with the fact those genes are highly expressed in the ovaries and accumulated in the oocytes. Also, it is interesting to note that the elevated expressions of VPS38/UVRAG and ATG14 in the tropharium suggest that the nurse cells (located in the tropharium and connected to the oocytes by cytoplasmic bridges) are indeed an important site of maternal mRNAs transcription and delivery to the

transcriptionally inactive developing oocytes [75], and this pattern has been previously observed for other ATGs in *R. prolixus* [76–78].

The silencing of VPS38/UVRAG and ATG14 resulted in altered biogenesis of the yolk organelles as observed in histology sections, and a shift to the formation of smaller and less granular organelles in the oocytes of silenced females could be observed using flow cytometry. It is possible that yolk organelles differences in contents may be detectable as changes in internal complexity. For example, the yolk proteins in Xenopus are packed in some yolk organelles in a crystalline structure [79], and internal vesicles and membranes have been observed inside some specific organelles in the oocytes of chicken and echinoderms [80, 81]. Because we do not know much about the nature of the distinct yolk organelles in *R. prolixus*, observing their overall pattern of relative size and internal complexity showed that indeed those organelles are diverse, and most importantly for this work, this pattern was altered in the oocytes produced by silenced females. There is no previous reference for this experiment or specific fluorescence labeling available for yolk organelles. Thus, it is not possible to rule out the detection of aggregates as typically performed in cell culture experiments using flow cytometry. Our lab intends to further explore this methodology and develop labels to detect specific yolk organelles under different conditions.

The phospholipid PI3P is the key signal generated by the PI3K complexes to recruit the specific effector proteins that promote endocytosis or autophagy [16, 82]. Because both VPS38/UVRAG and ATG14 are components of PI3K complexes, it was expected that their silencing would result in some decrease in the levels of PI3P, suggesting that the RNAi silencing of those genes resulted in at least some reduction of their protein activity. Plus, as previously observed for the silencing of ATG6/Beclin1, the lack of that signal molecule is likely the reason why silenced oocytes were not able to properly uptake the yolk proteins. The participation of endocytosis in the yolk uptake is not disputed and have been previously described in the oocytes of other insects [37–45]. Because PI3P is typically highly concentrated in early endosomes [83], and the mature yolk organelles are notably not formed in the oocytes of silenced females (Fig 5), it is likely that the silencing of the ATG6-PI3K complexes impair the endocytic flux at early steps of endosomal sorting. However, the identification of the different endocytic stages that could be compromised in the oocytes of silenced insects remains to be investigated.

The previous work from our group has shown that silencing of ATG6/Beclin1 is important for the yolk protein uptake, yolk organelles biogenesis and embryo viability, resulting in the accumulation of yolk proteins in the hemolymph and the production of small/white oocytes, which accumulated only 20% of the amount of yolk proteins detected in control oocytes [46]. Even though the ATG6/Beclin1 data indicated that the yolk endocytosis was compromised, we could not exclude the contribution of both complexes to the observed phenotypes, especially because the regulations encompassing the formation of each of the complexes are still unclear [15, 84]. To address this, in this work, we tested the silencing of the ATG6/Beclin1 PI3K complexes variant regulatory subunits: ATG14 (complex I, autophagy) and VPS38/UVRAG (complex II, endocytosis). Surprisingly, the silencing of both genes phenocopied the silencing of ATG6/Beclin1, impairing any additional phenotype-based interpretations on the contribution of each of the PI3K complexes to the previously observed compromised yolk accumulation. However, we found that the similar phenotypes were the result of a specific cross-silencing effect among the PI3K subunits VPS38/UVRAG, ATG14 and ATG6/Beclin1. Fig 6 shows that the silencing of VPS38/UVRAG and ATG6/Beclin1 resulted in the specific silencing of each other, whereas the silencing of ATG14 triggered the silencing of all three PI3K components. This data is interesting in the sense that it allows the speculation that the subunits of both PI3K complexes are under a circuit of gene regulatory network that shape the effects of PI3P production and signals during oogenesis. Gene regulation networks consist of transcription factors

and regulatory elements that altogether control spatiotemporal patterns of gene expression [85]. Furthermore, the fact that the expression of ATG14 is not altered by downregulation of VPS38/UVRAG and ATG6/Beclin1, allowed us to infer that the impaired yolk uptake phenotype (accumulated yolk protein in the hemolymph and the production of small/white oocytes and eggs), is, indeed, mostly the effect of silencing the VPS38 PI3K complex II, as in VPS38/UVRAG- and ATG6-silenced insects the very same phenotypes were reproduced without the cross silencing of ATG14. It is possible that the ATG14 PI3K complex is maternally loaded during oogenesis but activated only at later processes of oogenesis or early development such as egg activation, fertilization, and early yolk degradation. Activation of the autophagic flux after fertilization has been shown in mice, where it was associated with mRNA degradation for the maternal-zygotic transition [86]. In *C. elegans*, paternal mitochondria are degraded via autophagy after fertilization [87], and, in Drosophila, ATG mutants have varied phenotypes presenting compromised embryogenesis [88]. Yolk degradation, specifically, was associated with autophagic mechanisms in 2015 in Drosophila. In this work, the authors showed that target of rapamycin (TOR) and ATG1 are important for the yolk catabolism and for the formation of autophagosomes [89].

Considering that the three genes in question, and their predicted proteins, are quite divergent both in terms of sequence (the highest similarity value of 13,65% is between VPS38/UVRAG and ATG14) and function, and that all primers and dsRNA fragments were tested *in silico* for potential cross pairing with similar sequences against the *Rhodnius* transcriptome database, the possibility of primers and dsRNAs cross targeting to produce the cross-silencing effects is not supported. However, general off targets effects of RNAi have been described and cannot be ruled out in any RNAi silencing experiment [90–92]. For this reason, as a way to access if the cross-silencing regulation was specific for the subunits of the PI3K complexes, non-related genes were also tested, and, in general, showed no silencing tendency. Interestingly, however, the silencing of VPS38/UVRAG triggered the downregulation of ULK1/ATG1, and this did not occur in ATG14 or ATG6/Beclin1 silenced females, where VPS38/UVRAG was also cross-silenced. Because the crosstalk between endocytosis and autophagy is known and described in different models, it is possible that the signals for endocytosis also co-regulate the biogenesis of autophagosomes through the transcriptional regulation of ULK1/ATG1. The absence of the same effect in the samples where VPS38 was cross silenced (dsATG6/Beclin1 and dsATG14) might be due to the fact that gene regulatory networks are wired in a temporal regulated manner, so that the timing of the transcriptional downregulation was not enough to trigger the effect on ULK1/ATG1 observed when VPS38 itself was silenced. Experiments to test the levels of all three transcripts and the lipidation of ATG8/LC3 (as a readout of autophagosome formation) throughout the vitellogenesis period could help illuminate this hypothesis.

Altogether, we found the existence of a transcriptional regulatory system between the subunits of class-III PI3K complexes I and II, and that PI3P signals emerging from the class-III PI3K complexes are critical regulators of the yolk organelles biogenesis in the oocytes of the vector *Rhodnius prolixus*.

## Supporting information

**S1 Fig. 18S as a stable reference gene.** 18s Cts obtained from different samples and conditions. (A) RNA extracted from the ovary, midgut, fat body and flight muscle. (B) RNA extracted from the different parts of the ovariole (troph, tropharium; PV, previtellogenic follicle; Vit, Vitellogenic follicle; Chor, chorionated oocyte). (C) RNA extracted from the ovary and fat body of control (dsMal) and silenced insects. All samples were dissected 7 days after the blood meal.
(TIF)

**S2 Fig. Predicted introns and exons for the VPS38/UVRAG and ATG4 genes.** Sequence information was obtained from Vector Base (https://www.vectorbase.org/).
(TIF)

**S3 Fig. VPS38/UVRAG and ATG4 sequence and alignments with their orthologues from different species.** Reference sequence *Rp–Rhodnius prolixus. Dm–Drosophila melanogaster*; Hs–*Homo sapiens*; Sc—*Saccharomyces cerevisiae.*
(TIF)

**S4 Fig. Autoradiography of the TLC used to detect PI3P.** Control and silenced females were fed with a $^{32}$P-enriched blood meal. The chorionated oocytes were collected and subjected to lipid extraction followed by TLC and autoradiography. Arrows indicate the spots for PI3P and PI4P as previously described by [61].
(TIF)

**S5 Fig. RNAi knockdown of VPS38/UVRAG and ATG14 does not affect major physiological aspects of *R. prolixus*.** A. Effect of VPS38/UVRAG and ATG14 knockdown in the blood protein amount in the midgut of the females during digestion (n = 3). B. Survival curves of control and silenced females (n = 3). Graph shows mean ± SEM. p = 0.57 (VPS38) and p = 0.43 (ATG14), Log-rank (Mantel-Cox) test.
(TIF)

**S6 Fig. Hatching rates were monitored in eggs laid during the gonotrophic cycle of *Rhodnius prolixus*.** The major hatching phenotypes were observed in eggs produced during the second and third cycles of oviposition.
(TIF)

**S1 Table. Genes and primers List.** All sequences were obtained from *Vector Base* (https://www.vectorbase.org/) and primers were synthesized by Macrogen or IDT technologies.
(DOCX)

## Acknowledgments

The authors thank Yasmin Gutierrez, Mariana Martins and Yan Mendonça for the careful care of our lab *insectarium* and CENABIO-UFRJ and PIA-IBCCF/UFRJ for providing equipment and facilities.

## Author Contributions

**Conceptualization:** Priscila H. Vieira, Georgia Atella, Isabela Ramos.

**Data curation:** Priscila H. Vieira, Claudia F. Benjamim, Georgia Atella, Isabela Ramos.

**Formal analysis:** Priscila H. Vieira, Georgia Atella, Isabela Ramos.

**Funding acquisition:** Isabela Ramos.

**Methodology:** Priscila H. Vieira, Georgia Atella, Isabela Ramos.

**Project administration:** Isabela Ramos.

**Supervision:** Isabela Ramos.

**Writing – original draft:** Priscila H. Vieira.

**Writing – review & editing:** Claudia F. Benjamim, Georgia Atella, Isabela Ramos.

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
