## [Decision Letter · Decision Letter 0]

11 May 2021

Dear Dr Ramos,

Thank you very much for submitting your manuscript "VPS38/UVRAG and ATG14, the variant regulatory subunits of the ATG6/Beclin1-PI3K complexes, are crucial for the biogenesis of the yolk organelles and are transcriptionally regulated in the oocytes of the vector Rhodnius prolixus." for consideration at PLOS Neglected Tropical Diseases. As with all papers reviewed by the journal, your manuscript was reviewed by members of the editorial board and by two independent reviewers. In light of the reviews (below this email), we would like to invite the resubmission of a significantly-revised version that takes into account all the reviewers' comments. 

We cannot make any decision about publication until we have seen the revised manuscript and your response to the reviewers' comments. Your revised manuscript is also likely to be sent to reviewers for further evaluation.

Sincerely,

Alessandra Aparecida Guarneri

Associate Editor

Eric Dumonteil

Deputy Editor

Reviewer's Responses to Questions

**Key Review Criteria Required for Acceptance?**

**Methods**

-Are the objectives of the study clearly articulated with a clear testable hypothesis stated?

-Is the study design appropriate to address the stated objectives?

-Is the population clearly described and appropriate for the hypothesis being tested?

-Is the sample size sufficient to ensure adequate power to address the hypothesis being tested?

-Were correct statistical analysis used to support conclusions?

-Are there concerns about ethical or regulatory requirements being met?

Reviewer #1: The objectives of the study are clearly articulated with the hypothesis stated and the design is appropriate to address the stated objectives. However, important things or clarifications must to be address to ensure reproducibility of the results and credibility in the eyes of readers, in other words, a lot of information is missing, mainly in material and methods; that section is the heart of the article because it shows the validity of the work, so please check it and be sure you lead every suggestion (explaining or adding the missing information). 

1) Add the corresponding reference to R. prolixus genome and transcriptome databases, for example, Mesquita et al., 2015 or Giraldo-Calderón et al., 2015. 

2) Please, consider citing the reference to SIA software and PFAM.

3) GenBank/database accession numbers should be provided for all of the genes/transcripts covered in the manuscript (i.e. Drosophila, yeast and Human sequences used in in Fig 1). Also, the use of Yeast and Human sequences is missing in Material and Methods.

4) Insects section: are the insects also controlled by light and dark cycles? were the females mated? Were females in the first reproductive cycle used? In other words, after the first blood feeding as adult? What was the age of these females? Is the autogeny of these females controlled? All these questions should be answered in the manuscript to cover a rigorous methodology because the reproductive physiology in this specie, including the development of the oocytes, depends of all these parameters. 

5) The authors need to give more detail for the extraction methods, cDNA synthesis and qPCR (in what buffer the tissues were dissected? detail the cDNA synthesis reaction (amount of enzyme, reaction time and temperature, the primer used), detail the qPCR reaction (amount of primers, amount of cDNA (dilutions?), temperature cycle, use of blank without cDNA, detail the characteristics of the primers (add to Table 1: slope of the curve, intercept y and r2 of the cDNA curve), how ensure that treatment with DNAse I has degraded all possible contamination with genomic DNA?) The authors might find missing information that should be added using the MIQE.

6) For each experiment, did the authors use a pool of tissues or an individual tissue? Please, check it throughout the MS and add the information where necessary. 

7) How were the different parts of the ovariole obtained? Perhaps it could be necessary to define the different parts of an ovariole and how they were classified in this work to be dissected. Can the authors ensure that there is no contamination between the different parts? 

8) Also, in the Supplementary Fig 1 the author stated “different tissues of the ovary and oocyte stages” but it is actually different parts of an ovariole (not tissues). Please, check it throughout the MS.

9) In the supplementary Fig 1, it is stated that all samples were dissected 7 days after the blood meal but in Material and Methods section, it is reported that all organs were dissected 6 days after blood meal. Please, check it. I guess at 7 days PBM the egg laying has started, but again, it depends of all the condition asked in the 4th point. 

10) Did the authors separate follicle cells from oocytes? Otherwise, they must to be classified as follicles and not oocyte. Please, check it throughout the MS. Also, chorionated oocytes are actually eggs, right? So, how was “chorionated oocytes sample” defined if the egg laid has already started? I assume that just the eggs that are in the ovary but not in the oviduct. This is not very clear, please explain better. 

11) The RNAi experiments were done lack an off-target control. Without this control, off-target effects could not be ruled out. The authors made mention of this point in the discussion, but for future work, it would be interesting to have a second dsRNA with a different sequence to exclude this possibility, specially because mostly the conclusions got by this report are based on RNAi experiments. 

12) Why does the Table S1 show two pairs of primers per each transcript to RNAi experiments? Please, explain better the RNAi design. 

13) Where is the dsRNA injected? In what part of the body insect? Was the dsRNA diluted in water, PBS or Rhodnius solution? Please, add this information to the text.

14) Please, specify in which tissues the knockdown efficiency was confirmed.

15) Figure supplementary 2: Mortality curve is not analyzed with parametric statistics since it has a binomial distribution (dead or alive). An analysis using the Log-rank test would be more appropriate.

16) PI3P detection by thin-layer chromatography (TLC) and quantification: The TLC plates and the autoradiography of the phospholipids and standards used in this experiment should be show as Supplementary figure.

17) The densitometric analysis using Adobe Photoshop CC 2015 software might not be the best option, other applications such as ImageJ are more specific for this analysis. Otherwise, please give references to use Adobe Photoshop CC 2015 software for this analysis. 

18) In Material and methods, it is stated that at 10 days post blood meal, the “chorionated oocytes” were collected; however, in the Fig 2 caption, it is stated that the organs were collected at 7 d post feeding. Also, the caption of Fig 2 indicates that PI3P detection by TLC was done using ovaries; however, in Material and methods was pointed out that just “chorionated oocytes” were used for this experiment. 

19) Egg homogenates and SDS-PAGE: were the samples centrifugated or just homogenized and used? Please, give details of the method to homogenization and the recipe for PBS. 

20) Hemolymph extraction and SDS-PAGE: how was the hemolymph extraction done? Indicate the method used to obtain hemolymph. Was the hemolymph sample hemocyte-free? Please, indicate if the samples were centrifugated.

21) Did the authors use around 2 X more proteins for dsVPS38 or dsATG14 than dsMaI to load SDS-gel? It is confusing because an incorrect conclusion could be drawn from this experiment, indeed, the author conclusion is that the overall levels of protein in the hemolymph were increased in knockdown insects respect to controls, but it is complicated to analyze if the protein load for each condition was different. It could be concluded that yolk protein precursors are accumulated in hemolymph if the amount of protein load is the same; otherwise you might see more yolk protein because it loaded more. Please, check it or explain better. 

22) In Material and methods, it was written “dsRpATG6 samples”, but I think it was a mistake. 

23) “Hemolymphs” should be changed by “hemolymph samples”.

24) Light microscopy: were the tissues obtained only from opercula-free oocytes? Because in Material and methods only the preparation of this samples is indicated but then it is stated that transversal sections of the oocytes and eggs were obtained in a cryostat. Please, check it. What was the age of the females used for this experiment? How many days after blood meal? Please, indicate this relevant information in the text. 

25) Yolk organelles suspension and Flow Cytometry (FACS): The gating strategy for FACS analysis assessed by Flowing software to discriminate single yolk organelles from aggregate (doublets, clumps or debris) should be added as Supplementary Figure. Also, “PBS” should be abbreviated because it has already been used. What was the age of the females used for this experiment? How many days after blood meal? Please, indicate this relevant information in the text. 

26) Please clarify in Material and methods section the number of independent experiments, and the number of samples (individual or pools) composing each experimental group. This is valid for all the experiments described in the work.

Reviewer #2: The objective and hypothesis are well articulated. In general terms, the experimental designs are appropriate. Sample size is adequate to address the hypothesis. Statistical analyses were correctly applied.

**Results**

-Does the analysis presented match the analysis plan?

-Are the results clearly and completely presented?

-Are the figures (Tables, Images) of sufficient quality for clarity?

Reviewer #1: 1) Results: please, add as Supplementary information the alignment between the different sequences in order to demonstrate the percentage of similarity indicated in “Results”, as well as a picture showing the exons indicated by the authors for each transcript. 

2) Figure 1: indicate the statistical significance of the asterisk in the caption. 

3) The VPS38 and ATG14 transcript expression showed in Figure 1 is so different to that reported in Figure 2 (control insects). Please, explain this point. 

4) In Figure 2 A-B: is the complete ovary used for this experiment? 

5) Figure 2 C: please, be careful to discuss about if the complete ovary or just the chorionated oocytes have been used, it should not be used indistinctly. This is valid for all the manuscript.

6) Figure 2 C: It is not clear that silenced ovaries are presenting a reduction of 72% in their PI3P levels, it seems closer to 40%.

7) Fig 3E is not in Figure 3, please, check the text in results section.

8) It can’t be concluded from a simple SDS-gel that the bands noted as Vt/Vg and RHBP are exactly what the authors claim to be. We all know the molecular weights of them, but it would be necessary to indicate it with more care if you don’t use a specific antibody. Please, write in potential (the band is compatible with …) when indicating that these proteins could be Vt/Vg and RHBP, and refer to some other paper where the running profile by SDS-gel has been demonstrated for each protein.

9) The model for the transcriptional regulations between ATG6/Beclin1, VPS38/UVRAG and ATG14 is in Figure 6 but in the caption of Figure 5. Please, check it. 

10) Please, add the statistical significance missing in the caption of Figure 6.

11) Please, clarify if the level of knockdown presented (Knockdown (%) post-treatment) is lowering the expression by presented % or to presented %. Authors could add it in the caption.

12) Flow Cytometry (FACS): Is there any reference for this experiment? Otherwise, the authors should explain better what is the aim of measure the complexity of yolk organelles, because it is clear the reason to measure the forward direction (to indicate the relative size of the organelles), but the SSC is use to indicate the internal complexity or granularity of the cell, and here just yolk organelles are analyzed. Could the protein contained in each organelle (after discrimination single organelle from aggregates) give information about the complexity?

13) Figure 5: The Fig 5B doesn’t seem a representative density plots of the yolk organelles present in control and silenced chorionated oocytes if it is compared with the frequency showed on the upper quadrants (right and left, Fig 5C). In the Fig 5B it seems that in the RT, events in dsVPS38 RT are higher than control, and in the LT, events in both dsVPS38 and dsATG14 LT, are higher than control.

14) Figure 6: please, add in Material and methods section the use of dsATG6.

Reviewer #2: The analysis presented are in agreement with the analysis plan. The results are completely presented. Figures and Tables are of sufficient quality. Minor revision for specific Figures was indicated in "comments to authors".

**Conclusions**

-Are the conclusions supported by the data presented?

-Are the limitations of analysis clearly described?

-Do the authors discuss how these data can be helpful to advance our understanding of the topic under study?

-Is public health relevance addressed?

Reviewer #1: 1) Discussion: it seems that a word is missing in the first sentence. Please, check it. 

2) Yolk protein precursors are accumulated in coated vesicles and then, after loosing clathrin coats, in early endosomes. These early endosomes then fuse and form the yolk organelles; in this context, it would be great to discuss what of these potential levels were affected.

3) The fact that the expression of ATG14 is not altered by downregulation of VPS38/UVRAG and ATG6/Beclin1 allows to conclude that the impaired yolk uptake phenotype by dsATG6 treated females is the effect of silencing the VPS38 PI3K complex II, which could indicate the VPS38/UVRAG PI3K complex II as the major contributor just for yolk endocytosis, and then, could ATG14 (complex I) be working on further steps, e.g. during the process of protein degradation (autophagy)?

Reviewer #2: Most of conclusions are supported by the findings, but discussion needs to be improved. Please, see “comments to authors” for details.

**Editorial and Data Presentation Modifications?**

Reviewer #1: Introduction: 

1) Since the scope of PLOS Neglected Tropical Diseases is aimed to report studies showing aspects of forgotten diseases affecting the world’s most neglected people, information about the relevance of this work in the current context of Chagas disease should be added, specially because since 2005, the WHO recognized this affection as a neglected tropical disease (NTDs), which is mainly transmitted by the contact of people with faeces/urine of infected blood-sucking triatomine bugs.

2) Also, I realized that a good background talking about endocytotic organelles in insect oocytes, or showing the nature of yolk bodies, was missing; it should be added in the introduction to show the relevance of the complexes studied here, in the regulation of autophagy and membrane dynamic of endocytosis.

3) Could the authors check the references, and perhaps use more reviews when possible? for example, at the end of a sentence in the introduction there are 14 references added; I understand that it is also part of previous stated, but in that case, it would be better to use after each sentence the corresponding reference.

4) ATG9 is named in the 3rd paragraph, I suggest that each time something new is named in the text, it is briefly defined. 

5) It may be helpful for the reader to indicate the full name of the abbreviation the first time they are named, for example, UV radiation resistance-associated gene protein (UVRAG) Vacuolar protein sorting 34 (Vps34), AuTophaGy-related (ATG), etc.

Reviewer #2: Please, see Summary and General Comments.

**Summary and General Comments**

Reviewer #1: In this study, the authors performed an analysis on the biogenesis of yolk organelles in the oocytes of the vector Rhodnius prolixus, delving into a study previously published by the same group. Here they report that VPS38/UVRAG and ATG14 subunits of the ATG6/Beclin1-PI3K complexes are essential for the yolk endocytosis. The work is original in the study area on biogenesis of yolk bodies and the conclusions are justified by the evidence presented. Overall, I consider it work contains very interesting and relevant data and deserves to be published. However, it is essential to address a significant number of points to improve the understanding and reproducibility of the experiments. Also, I want to state that the page numbers and line numbers in the manuscript file were missing, so it has been a bit complicated to indicate my observations.

Reviewer #2: Comments to the authors:

In this work, Veira and co-authors analyzed the role of the variant subunits VPS38/UVRAG and ATG14 in the biogenesis of the yolk bodies in the insect vector of Chagas Disease Rhodnius prolixus. Both genes are members of the ATG6/Beclin1 class-III PI3K complexes, which are involved in the regulation of autophagy and the membrane dynamics of endocytosis. The aims are clear and straightforward and the results are compelling. Even though some conclusions must be revised, the manuscript open horizons to unravel the regulating processes involved in protein uptake and trafficking by developing oocytes, which are largely unknown. Please, find below some points that I suggest revising in order to improve this manuscript. 

General Comments:

- Through the manuscript: Sentences like “control and silenced eggs …” or “chorionated oocytes..” are incorrect. Samples were obtained from control and silenced females. Please, revise. 

- Material and Methods, page 10- Yolk organelles suspension and Flow Cytometry (FACS): Consider replacing “Suspensions of yolk organelles were obtained by gently disrupting recently dissected chorionated…” by “Suspensions of yolk organelles were obtained by gently disrupting of recently dissected chorionated…”

- Results, page 10-Silencing of VPS38/UVRAG and ATG14 results in the accumulation of yolk proteins in the hemolymph and formation of small yolk-deficient oocytes: Consider replacing “Double stranded RNAs were designed to target specific regions of the VPS38/UVRAG and ATG14 mRNAs and injected to the insects hemocoel 2 days before the blood meal” by “Double stranded RNAs designed to target specific regions of the VPS38/UVRAG and ATG14 mRNAs were injected to the insects hemocoel 2 days before the blood meal.

- Results, page 11- VPS38/UVRAG and ATG14 silenced eggs are yolk-deficient and do not support embryo development: Consider replacing “… the silenced females laid the same number of eggs as control females”…by “… the silenced females laid similar number of eggs as control females”…

Specific Comments

a) Results and Figures:

- Silencing of VPS38/UVRAG and ATG14 results in the accumulation of yolk proteins in the hemolymph and formation of small yolk-deficient oocytes (page 11): The authors proposed the measurement of hemolymph yolk protein content in control and silenced females to demonstrate that formation of white/small oocytes was due to a deficiency in the uptake of yolk proteins from the hemolymph. Such an assumption is reasonable but not conclusive if information regarding expression (mRNA and protein levels) of yolk protein precursors such as vitellogenin in the fat body is not provided. In addition, the results showed the amount of total proteins in the hemolymph but not yolk protein content. Authors should perform at least western blots for vitellogenin and RHBP in fat body to better support the conclusion. Please, limit the rationale of the experimental design.

-VPS38/UVRAG, ATG14 and ATG6/Beclin1 (page 12) and Figure 6: Authors should provide a hypothesis to explain the downregulation of ULK1/ATG1 in VPS38/UVRAG-silenced samples.

- Fig. 3D and Fig. 4F: Did the authors use a loading control for SDS-PAGE? According to Material and Methods section, the amount of proteins loaded were different in samples from control and silenced females.

- Legends to Figure 5 and Figure 6 (page 18): The model for the transcriptional regulations between ATG6/Beclin1, VPS38/UVRAG and ATG14 is shown in Figure 6, not in Figure 5. Please, revise. 

- Supplementary Figure S2: Survival curves are presented in the panel “B”, but the legend stated “E-F”. Please, revise.

b) Discussion:

- Authors associated the decrease of PI3P in ovaries/chorionated oocytes from VPS38/UVRAG and ATG14 silenced females with “at least some reduction of their protein activity, validating the specificity of the observed phenotypes”. In fact, signaling mediated by PI3P is complex and can be performed even with low levels of PI3P. Please, tone down the conclusion. 

- Authors scarcely compare their results with others in the literature from different species. Please complete.

PLOS authors have the option to publish the peer review history of their article (what does this mean?). If published, this will include your full peer review and any attached files.

Reviewer #1: No

Reviewer #2: No
---

## [Decision Letter · Decision Letter 1]

24 Aug 2021

Dear Dr Ramos,

We are pleased to inform you that your manuscript 'VPS38/UVRAG and ATG14, the variant regulatory subunits of the ATG6/Beclin1-PI3K complexes, are crucial for the biogenesis of the yolk organelles and are transcriptionally regulated in the oocytes of the vector Rhodnius prolixus.' has been provisionally accepted for publication in PLOS Neglected Tropical Diseases.

Best regards,

Alessandra Aparecida Guarneri

Associate Editor

Eric Dumonteil

Deputy Editor

Reviewer's Responses to Questions

**Key Review Criteria Required for Acceptance?**

**Methods**

-Are the objectives of the study clearly articulated with a clear testable hypothesis stated?

-Is the study design appropriate to address the stated objectives?

-Is the population clearly described and appropriate for the hypothesis being tested?

-Is the sample size sufficient to ensure adequate power to address the hypothesis being tested?

-Were correct statistical analysis used to support conclusions?

-Are there concerns about ethical or regulatory requirements being met?

Reviewer #1: yes

Reviewer #2: As stated in the first revision, the objective and hypothesis of the MS are well interconnected. The experimental designs are appropriate. Sample size is adequate to address the hypothesis. Statistical analyses were correctly applied.

**Results**

-Does the analysis presented match the analysis plan?

-Are the results clearly and completely presented?

-Are the figures (Tables, Images) of sufficient quality for clarity?

Reviewer #1: yes

Reviewer #2: As stated in the first revision, the analysis presented are in agreement with the analysis plan. The results are completely presented. Figures and Tables meet appropriate quality standards.

**Conclusions**

-Are the conclusions supported by the data presented?

-Are the limitations of analysis clearly described?

-Do the authors discuss how these data can be helpful to advance our understanding of the topic under study?

-Is public health relevance addressed?

Reviewer #1: yes

Reviewer #2: Authors made suitable modifications to strengthen the conclusions based on the obtained results. According with the suggestions, the discussion section was largely improved.

**Editorial and Data Presentation Modifications?**

Reviewer #1: -

Reviewer #2: (No Response)

**Summary and General Comments**

Reviewer #1: The authors have addressed all comments in the revised MS. This is a nice work. Congratulations.

Reviewer #2: Comments for the authors:

The authors have considered most of the suggestions made by this reviewer. Other aspects related to the methodology and results have also been satisfactorily answered. The authors conducted an exhaustive revision of the entire manuscript, including review of the English language.

The revised version of the manuscript has improved substantially.

PLOS authors have the option to publish the peer review history of their article (what does this mean?). If published, this will include your full peer review and any attached files.

Reviewer #1: No

Reviewer #2: No

---

## [Editor Report · Acceptance letter]

31 Aug 2021

Dear Dr Ramos,

We are delighted to inform you that your manuscript, "VPS38/UVRAG and ATG14, the variant regulatory subunits of the ATG6/Beclin1-PI3K complexes, are crucial for the biogenesis of the yolk organelles and are transcriptionally regulated in the oocytes of the vector Rhodnius prolixus.," has been formally accepted for publication in PLOS Neglected Tropical Diseases.

Best regards,

Shaden Kamhawi

co-Editor-in-Chief

Paul Brindley

co-Editor-in-Chief
